# The Inhibitory Mechanism of 7*H*-Pyrrolo[2,3-d]pyrimidine Derivatives as Inhibitors of P21-Activated Kinase 4 through Molecular Dynamics Simulation

**DOI:** 10.3390/molecules28010413

**Published:** 2023-01-03

**Authors:** Juan Du, Song Wang, Xinyue Zhang, Chang Liu, Yurou Zhang, Hao Zhang

**Affiliations:** Institute of Theoretical Chemistry, College of Chemistry, Jilin University, Changchun 130023, China

**Keywords:** PAK4, 7*H*-pyrrolo[2,3-d]pyrimidine, molecular dynamics simulation, MM/PBSA, inhibitory mechanism

## Abstract

The overexpression of p21-activated kinase 4 (PAK4) is associated with a variety of cancers. In this paper, the binding modes and inhibitory mechanisms of four 7*H*-pyrrolo[2,3-d]pyrimidine competitive inhibitors of PAK4 were investigated at the molecular level, mainly using molecular dynamics simulations and binding free energy calculations. The results show that the inhibitors had strong interactions with the hinge region, the β-sheets, and the residues with charged side chains around the 4-substituent. The terminal amino group of the inhibitor 5n was different from the other three, which could cause the enhancement of hydrogen bonds or electrostatic interactions formed with the surrounding residues. Thus, inhibitor 5n had the strongest inhibition capacity. The different halogen atoms on the 2-substituents of the inhibitors 5h, 5g, and 5e caused differences in the positions of the 2-benzene rings and affected the interactions of the hinge region. It also affected to some extent the orientations of the 4-imino groups and consequently their affinities for the surrounding charged residues. The combined results lead to the weakest inhibitory capacity of inhibitor 5e.

## 1. Introduction

The p21-activated kinases (PAKs) are serine/threonine protein kinases that are widely found in eukaryotes [1]. The current study found that the mammalian PAK family contains six members [2], namely PAK1-6 [3]. PAK4 is one of the PAKs most closely associated with cancer [3]. It is the key effector in a variety of signaling pathways, transmitting to downstream factors both inside and outside the cell and involved in processes such as cell growth, apoptosis prevention, cell proliferation, and senescence regulation [4,5]. Many scientific studies confirmed that PAK4 is overexpressed in a variety of human cancers, such as breast [6], pancreatic [7], gallbladder [8], gastric [9], hepatocellular [10], and esophageal cancers [11].

PAK4 has the common structure of protein kinases, containing an N-terminal regulatory domain and a C-terminal kinase domain [12]. The C-terminal kinase domain of PAK4 catalyzes γ-phosphate transfer from ATP to the receptor polypeptide [12]. Its kinase domain is shown in Figure 1, with residues ranging from Ser300 to Arg591 [13]. It contains two kinase lobes [14], the N-lobe consisting mainly of five antiparallel aligned β-sheets (β1-β5) and three α-helices (αA-αC), and the C-lobe consisting mainly of α-helices (αD-αJ) [12,15]. The two kinase lobes are connected by a hinge, and a larger cleft between the two lobes is used to bind the phosphate donor ATP [14]. Previous studies suggested that the regions associated with catalytic activity include the glycine-rich loop (P-loop, Gly328-Gly333), helix αC (Arg360-Met370), hinge region (Glu396-Gly400), catalytic loop (His438-Ser445), and activation loop (T-loop, Asp458-Val476) [14,16].

Studies in recent years showed that PAK4 inhibitors can inhibit its expression, thus preventing tumor growth, inducing tumor regression, and avoiding tumor cell metastasis, which has a good effect on the treatment of cancers [17]. PAK4 inhibitors can be divided into ATP-competitive inhibitors and non-competitive allosteric inhibitors depending on their binding positions and modes of action [18]. Among them, the study on the pyrrolopyrazole inhibitor PF-3758309 is relatively early. PF-3758309 is an ATP-competitive inhibitor with high inhibitory activity against PAK4. It was in phase I clinical trials but was eventually stopped due to low oral bioavailability and poor permeability [19,20]. PKT-9274, a representative allosteric inhibitor, was used to treat solid tumors and non-Hodgkin’s lymphoma in phase I clinical trials [21]. 

Pyrrolopyrimidine alkaloids attracted the attention of medicinal chemistry researchers since the beginning of isolation and synthesis [22]. Pyrrolo [2,3-d] pyrimidine is an important part of many biologically active compounds and is a common motif in a number of natural products and drugs [22]. It was found that pyrrolo [2,3-d] pyrimidine has anticancer [23], antiviral [24], antibacterial [25], and enzyme-inhibiting [26] properties. Recently, Wang et al. at China Pharmaceutical University studied a series of ATP-competitive inhibitors, 7*H*-pyrrolo[2,3-d]pyrimidine derivatives, which showed good inhibitory capacity for PAK4 with no cytotoxicity [26]. This class of inhibitors has two different substituents connected to the pyrrolo [2,3-d] pyrimidine parent nucleus, and the differences in substituents can result in a 1000-fold or more difference in inhibitory capacity. They also explored the enzyme-inhibitor binding modes using a molecular docking method [26]. However, docking calculations can only qualitatively study enzyme-inhibitor binding patterns. The protein is set as rigid in their docking calculation, and the conformation cannot be adjusted according to the structures of different inhibitors, and the quantitative binding free energy cannot be given. This is not sufficient for competitive inhibitors whose inhibitory capacities are mainly affected by affinity. In this work, four representative inhibitors of pyrrole [2,3-d] pyrimidine derivatives were selected as research targets (their structures, group names, and IC_50_ values are shown in Figure 2) [26]. The binding modes between PAK4 and four different inhibitors were studied by methods of molecular docking, molecular dynamics simulation, and binding free energy calculation, and the reasons for the different inhibitory capacities were speculated. We first chose inhibitor 5n and 5e, which have the greatest difference in inhibitory capacity. Since inhibitor 5n only has a measured IC_50_ value, when selecting other inhibitors, we also chose inhibitors with a measured IC_50_ rather than kinetic activity. In addition, we tried to avoid selecting isomers. Similar methods were used by Guo et al. to study the binding patterns of PAK4 and other types of inhibitors, and detailed and quantitative results were obtained [27,28,29]. The four inhibitors we selected can be divided into two comparison groups to study the effects of different 2-substituents and 4-substituents on inhibitory capability. The first comparison group was inhibitors 5n and 5h, which have the same 2-substituents but different 4-substituents. By comparing them, we can study the effect of different 4-substituents on inhibitory capability. The second comparison group was the inhibitors 5h, 5g, and 5e, which have the same 4-substituents and different 2-substituents. By comparing them, the effect of different 2-substituents on inhibitory capability can be studied. Our study will provide some guidance for the design of new inhibitors.

## 2. Results and Discussion

### 2.1. Acquisition of PAK4 Protein and Inhibitor Structures Figures

The X-ray crystal structure of PAK4 (PDB code: 5XVG, resolution: 2.10 Å) can be obtained from the RCSB Protein Data Bank (http://www.rcsb.org/pdb, accessed on 15 December 2022) [30]. The solvent molecules and the original inhibitor in the crystallization were removed to obtain PAK4 apoprotein. The structures of the inhibitors were obtained from the PubChem chemical database [31] (the inhibitor 5h was not available from the database, so GaussView6.0.16 was used to construct the inhibitor 5h by referring to other structures). All four inhibitors were optimized at the B3LYP/6-31G(d) [32] level using the Gaussian 16 package to obtain stable structures and molecular parameters.

### 2.2. Molecular Docking Results

We used the AutoDock Vina software [33,34,35] to dock each of the four inhibitors into the PAK4 apoprotein. As the inhibitors we studied are all ATP-competitive inhibitors, the inhibitors bind primarily in the ATP-binding pocket. The ATP binding site was given in another crystal structure (PDB code: 4XBR) [16]. The center of the docking box could be determined by referring to the binding site of ATP in 4XBR. We selected the docking results from the lower free energy conformations that were similar to the crystal structure and the binding mode previously reported in the literature as the initial conformations. The results of the docking show that all four inhibitors had similar binding postures. The affinities of inhibitors 5n, 5h, 5g, and 5e with PAK4 were −36.784, −35.530, −35.530, and −34.694 kJ·mol^−1^, respectively. All four inhibitors formed double hydrogen bonds with Leu398 in PAK4 and one hydrogen bond with Glu396 and Asp444. In addition to the shared hydrogen bonds, the inhibitor 5n formed a hydrogen bond with Glu329; 5h formed a hydrogen bond with Asp458; and 5g formed a hydrogen bond with Ala402. The four docking results were constructed into complexes with PAK4 apoprotein, named complex 5n, complex 5h, complex 5g, and complex 5e, respectively.

### 2.3. Molecular Dynamics Simulations, Stability Analysis and Protein Flexibility Analysis

Using the complex structures obtained by molecular docking as the initial conformations, we performed molecular dynamics simulations of all the complexes using the GROMACS 2019.6 program [36,37,38]. As a reference, the same simulation was performed for PAK4 apoprotein. After 200 ns of product simulations, we obtained the kinetic trajectories of the PAK4 apoprotein and the four complex systems. Figure 3 depicts the variation curve of the root mean square deviation (RMSD) of the Cα atom relative to the initial position of the proteins in the five systems with simulation time. The RMSD of all five systems reached an acceptable equilibrium (fluctuations of less than 0.1 nm) after 50 ns. The mean values of the RMSD of the four complexes throughout the simulations were 0.139, 0.188, 0.158, and 0.181 nm, respectively. Since we chose the same reference conformation in the calculation of RMSD, the larger the RMSD difference is, the more the protein deviates from the reference conformation, which may cause changes on the protein surface. The mean solvent accessible surface area (SASA) values for the four complexes were 146.3, 147.3, 145.0, and 147.5 nm^2^, respectively. Although there was no strict correspondence in order with the RMSD, the general trend was consistent with complex 5n and 5g being smaller than complex 5h and 5e.

The root–mean–square fluctuation (RMSF) can be used to assess the flexibility of the protein. When a protein binds to an inhibitor, the region where it interacts with the inhibitor exhibits reduced flexibility (i.e., reduced RMSF) due to restriction. The comparisons between the RMSF of each residue in the four enzyme-inhibitor complex systems we simulated and the PAK4 apoprotein are shown in Figure 4. As can be seen from the figure, the RMSFs of all complexes decreased significantly compared with PAK4 apoprotein in the following areas:

The RMSFs of the four complexes did not differ significantly in Asp322-Gly333, which primarily contained β1 sheet and P-loop, but complex 5n/5h was significantly lower than complex 5g/5e in the P-loop region (Gly328-Gly333). The RMSFs of the four complexes did not differ significantly in the Arg341-Lys345 region, which was the loop connecting the β2 and β3 sheets. The Lys350-Glu366 region mainly contained a partial β3 sheet and helix αC. The RMSFs of complex 5n/5h in this region were significantly higher than that of complex 5g/5e, which might be related to the fact that the helix αC in the first two complexes was easy to unwind to a certain extent, which will be discussed in more detail in the following chapters. The differences in RMSF in the three regions mentioned above in different systems indicated that the inhibitors bound more tightly to the β1-β3 sheets, affecting the flexibility of the regions near these sheets. The Met395-Ala402 region was located on the hinge region reported in the literature that is most important for binding to ATP or inhibitors [13]. All four complexes showed a slight decrease in RMSF values in this region compared to PAK4 apoprotein. However, this region was a rigid region with a low RMSF compared to other regions with high flexibility variation, and therefore its RMSF value was not significantly reduced. Arg472-Met482 is a region on the T-loop. According to the literature, there is a strong interaction between the T-loop and the helix αC [14], and differences in the flexibility of this region in different systems might be related to helix αC changes. The region, Pro514-Ser542, was far from the inhibitor, and its RMSF change might be primarily influenced by the solvent.

### 2.4. Free Energy Landscape and Sampling

A projection of the MD simulation trajectories onto the first two eigenvectors (PC1 and PC2) gave a free energy landscape (FEL), which reflected the changes in conformational energies in the simulation trajectories [39,40]. The FELs of the four systems studied in this paper are shown in Figure 5, where the darker the blue, the lower the corresponding conformational energy. We extracted two conformations with lower energies on the free energy landscapes from the stable RMSD of each complex as samples for further study. Among them, the first group was the main research object of conformational analysis, and the corresponding sampling time in the four complex systems was 187.77 ns, 162.54 ns, 165.24 ns, and 167.26 ns, respectively. All conformational features observed in the first set of samples were confirmed in the second set of samples (sampling times of 117.39 ns, 109.35 ns, 123.81 ns, and 110.52 ns, respectively). The 3D conformations of the first set of samples can be found in the Appendix A.

### 2.5. Conformational Analysis of Samples

The conformational overlap of enzymes and inhibitors in the four complex samples we extracted is shown in Figure 6. As can be seen from the figure, there was little difference in conformations between the parent nucleus and the 2-benzene ring of the inhibitors in different complexes, with only slight translation. In contrast, the entire 4-substituent group rotated easily. In each of the four complexes, the 4-imino and 4-hexamoric ring were oriented differently. The overall conformations of the proteins were not significantly different, but some regions of the proteins had more pronounced differences in conformation. In all complexes, the helix αC and αEF (a short helix between αE and αF) showed varying degrees of unwinding. The helix αG had no unwinding, but the position varied greatly. The C-terminal of the T-loop, the P-loop, and the loop connected the β3 sheet to the helix αC—these three regions also had more pronounced conformational differences. The above-mentioned regions were all regions in RMSF that showed flexible changes. Furthermore, a 3-helix on the catalytic loop of complex 5n unwound, as did helices αA and αB in complex 5e.

### 2.6. Secondary Structure Analysis

The differences in secondary structures observed in different complexes during conformational analysis were individual phenomena in the samples. To verify that these phenomena were prevalent throughout the simulations, we performed secondary structure analysis of the proteins in the simulated trajectories using the do_dssp tool in GROMACS (as shown in Appendix A). As you can see from the figure, none of the β-sheets showed damage. The helix αG, with only a positional change in the sample conformation comparison, remained relatively intact throughout the simulations. Helix αC was the most obvious unwinding and was significantly different among the four complexes. The unwinding of the 3-helix on the catalytic loop of complex 5n and the unwinding helix αB of complex 5e were shown in the conformational observation of the samples. This phenomenon was observed in all complexes, but in different proportions. The unwinding of helix αA was indeed significant only in complex 5e. The Pro479-Trp481 segment of Helix αEF showed frequent transitions between 3-Helix and Turn, but in different proportions among the four complexes. Similar changes were observed in the Ser443–Ser445 segment of the catalytic loop.

### 2.7. Hydrogen Bonding Analysis

To investigate the formation of hydrogen bonds between inhibitors and proteins, we counted the average numbers of hydrogen bonds in all 20,000 snapshots of each system as well as the percentages of different residues forming hydrogen bonds with the inhibitors (as listed in Appendix A). As can be seen from the table, the average numbers of hydrogen bonds of the four complexes were not very different, and the complex 5n with the most hydrogen bonds had only 0.278 more hydrogen bonds than the complex 5g with the least. In all complexes, the -NH- and -CO- groups on the main chain of hinge region residues Leu398 formed double hydrogen bonds with the 1-N atom and the 2-imino group on the parent nucleus in the inhibitors (hence the presence of hydrogen bonds at this residue is greater than 100%). The carbonyl oxygen on the peptide plane between Glu396 and Phe397 formed a large proportion of hydrogen bonds (50% to 65%) with the parent nucleus 7-NH, suggesting that the hinge region was stably bound to the inhibitors mainly through approximately parallel triple hydrogen bonds (as shown in Figure 7a). In complex 5n, Ile327 formed a proportion of hydrogen bonds (14.43%) with the terminal amino group of the inhibitors. The lower occurrence of this hydrogen bond was mainly due to the fact that the distance between the bonding atoms (an average distance of 0.42 nm) was slightly longer than the hydrogen bonding range set in the automated software statistics (0.35 nm). In other complexes, the terminal amino group of the inhibitors became a terminal imino group, and Ile327 was much further away from the terminal imino group, so it had a low occurrence of hydrogen bonds (as shown in Figure 7b,c). Asp444 in complex 5n and Glu329 in complex 5e could also form a small number of hydrogen bonds with terminal amino groups, and their average distances between bonding atoms were farther than that of Ile327 (as shown in Figure 7b,c), so the occurrences of hydrogen bonds were lower (6–8%). As can be seen above, complex 5n had a slightly higher average number of hydrogen bonds than the other three complexes, mainly because the inhibitor 5n had a characteristic terminal amino group that could form more hydrogen bonds with residues such as Ile327 and Asp444.

### 2.8. Binding Free Energy Analysis

The four inhibitors studied in this paper were all ATP-competitive inhibitors, and their inhibitory capacities were closely related to the affinities between the inhibitors and proteins. We calculated the binding free energy of four inhibitors to PAK4. The 20 ns with relatively small conformational changes were truncated from the molecular dynamics simulation trajectories of the four complex systems, and then the g_mmpbsa tool was used to calculate the binding free energy of proteins and inhibitors in each system. Complex 5n was selected as the reference zero, and the absolute values of ΔG_bind_ of all complexes were changed to relative values.

The relative contributions of each branch of protein-inhibitor binding energy are listed in Appendix A. The relative ΔG_bind_ of complex 5n, 5h, 5g, and 5e were 0, 13.752, 19.803, and 35.162 kJ·mol^−1^, respectively. This was qualitatively consistent with the inhibitory order of the inhibitors. The relative solvation free energies (ΔG_PB_ + ΔG_SA_) in the four complexes were 0, 2.341, −9.455, and −5.528 kJ·mol^−1^, respectively. It can be seen that the differences in the solvation energies of complex 5n compared to 5h, as well as complex 5g compared to 5e, were small and accounted for a low proportion of the total ΔG_bind_ difference, except in complex 5h and 5g, where the difference in solvation free energy was larger. This difference might be related to the large difference in SASA between complex 5h and complex 5g that we mentioned before.

To identify the key residues in the proteins that interact with the inhibitors, we then calculated the decomposition contribution of each residue to the total binding free energy. The E_MM_ values of residues with large contributions are listed in Table 1.

As can be seen from the table, the residues with the largest E_MM_ contributions in all complexes were Glu396, Phe397, and Leu398, located on the hinge region. Of these, both Glu396 and Leu398 had a large occurrence of hydrogen bonds to the parent nucleus or 2-imino group in the hydrogen bonding statistics, and it was this strong interaction that made the parent nucleus and 2-substituent less susceptible to flipping. This was consistent with what was previously reported in the literature [13,26,28,41]. However, the interactions between these two residues were very strong in all inhibitors, suggesting that they were the main reason for the inhibitory capacities of the inhibitors but not necessarily the main reason for the difference in the inhibitory capacities of the different inhibitors. The atoms of both residues forming hydrogen bonds were carbonyl oxygens and amino groups in the peptide plane between Phe397. Generally, when forming these hydrogen bonds, the adjacent Phe397 also showed a larger E_MM_ contribution. In addition, there might be strong stacking interactions between the side chain benzene ring of Phe397 and the 2-benzene ring of the inhibitors. Gly401, which was also located on the hinge, showed strong E_MM_ contributions in all four complexes. Its main chain oxygen atom was close to 2-imino and might have some electrostatic interaction with it (as shown in Figure 8). The average distance between the main chain oxygen of Gly401 and the nitrogen atom on the 2-imino was significantly greater in complex 5g (0.664 nm) than in the other three complexes (0.562 nm, 0.551 nm, and 0.596 nm). Accordingly, the E_MM_ contribution of Gly401 in complex 5g was also significantly smaller than that in the other three complexes, which qualitatively confirmed the main type of Gly401 interactions with the inhibitors.

As shown in Figure 9a, there were three parallel sheets of β3, β2, and β1 above the binding position of the inhibitors. There were residues with large E_MM_ contributions on the three β-sheets. Of these, Ala348 was located on the β3 sheet, it and Leu447 on the opposite β7 sheet clamped the parent nucleus from both sides of the conjugated plane. Their hydrophobic side chains formed a sandwich structure with the parent nucleus, showing strong hydrophobic effects and a certain intensity of E_MM_ contributions. Val335 and Ile337 were located on the β2 sheet, and they linked the nearly 4-hexamoric ring and the 2-benzene ring by the hydrophobic side, respectively, which might form certain hydrophobic interactions. The E_MM_ contributions suggested that the Ile337 interaction with the inhibitors were significantly stronger in complex 5g than in the other three complexes. Conformational observation showed that this residue was close to the halogen atom on the 2-benzene ring of the inhibitors (as shown in Figure 9b). In complex 5g, this halogen atom was F. The F atom on the benzene ring had a more negative charge, and the F atom could form some interactions with the hydrogen on the polarized C-H bond. Thus, the stronger E_MM_ of Ile337 in complex 5g might be related to the F atom in the inhibitor. Ile327 was on the β1 sheet, and its side chain was located between the 2-benzene ring and the 4-hexamoric ring, possibly forming hydrophobic interactions with these two groups (as shown in Figure 7b,c). In addition to hydrophobic interactions, the main chain oxygens of Ile327 had small proportions of hydrogen bonds to the terminal amino or imino groups in all four complexes, as described above. In complex 5n, the hydrogen bond occupancy rate can even reach 14.43% due to its characteristic terminal amino group. Therefore, the E_MM_ contribution of this residue was the largest in complex 5n.

The residues Lys350, Glu366, Asp405, Asp444, Asp458, Arg589, and Arg591 all had charged side chains and exhibited significant E_MM_ contributions. They might form electrostatic interactions with the charged groups in the inhibitors. 

Lys350, Glu366, Asp405, Arg589, and Arg591 all had significantly different E_MM_ contributions in complex 5h than the other three. We speculated that they interacted with the same charged group in the inhibitors and that this charge group was more variable in the complex 5h. The conformational comparison of the inhibitors showed that the orientation of 4-imino in complex 5h was opposite to that in the other three complexes (as shown in Figure 10a,b). Lys350 and Glu366 were on the same side of this imino and formed salt bridges between the two, which together formed electrostatic interactions with the imino, where Lys350 exhibited repulsive forces and Glu366 exhibited gravitational forces. Asp405 was on the other side of the imino and formed an electrostatic interaction with it alone. The amino group of complex 5h flipped, and the negatively charged N atom and positively charged H atom were facing opposite directions. This resulted in a large change in the electrostatic interactions and led to the E_MM_ values of the above three residues in the complex being significantly different from those in the other three complexes. The side chains of residues Arg589 and Arg591 were positively charged, forming an attraction between the 2-benzene ring and the 4-hexatomic ring and forming a repulsive force with the positively charged hydrogen atom on 4-imino and cancelling part of the attraction. These two residues were located in the C-terminus of the protein and were more affected by the random influence of the solvent. In complex 5h, when it moved randomly to the side of the inhibitor, it tended to stabilize in this region due to the greater attractive force. However, in the other complexes, the attraction was too small to be easily stabilized. Therefore, these two residues also showed a large conformational difference in complex 5h and in the other complexes. To ensure the flip of 4-imino was not an individual phenomenon in the sample conformation, we also counted the dihedral angles that were most sensitive to this flip (the four atoms forming the dihedral are marked in Figure 10a,b, and the value of this dihedral angle during the simulation is shown in Figure 11). As can be seen from Figure 11, this dihedral angle in complex 5g did not flip during the entire simulation. In complex 5n and 5e, this dihedral angle, although also flipped, was not flipped in as large a proportion as in complex 5h. The orientation change in 4-imino might be the result of the combined actions of the charged groups on the 2-benzene ring and the 4-hexamoric ring on either side. The effect of the 4-hexamoric ring was more complex and remains unclear for the time being. However, the effects of the halogen atoms on the 2-benzene rings were more clearly seen when comparing 5h, 5g, and 5e, which had the same 4-hexamoric rings. As shown in Figure 12, the substitution positions of the halogen atoms on the 2-benzene ring were very close to 4-imino. When it was a negatively charged F atom, the positively charged hydrogen atom on the 4-imino had a strong electrostatic interaction with it and was not easily flipped. Therefore, in complex 5g, the 4-imino group did not flip during the simulation process. When it was an H atom (inhibitor 5e), it had less effect on the 4-imino, which could be flipped but in a lower proportion. When it was a Cl atom (inhibitor 5h), it could form a hydrophobic region around it to repel the hydrophilic imino due to the lesser charge on the Cl atom and the stronger hydrophobic effect, so a greater proportion of the imino flipped to the side away from the Cl atom. 

Both Asp444 and Asp458 were located near the 4-substituent group of the inhibitors. They formed electrostatic interactions with the charged groups on the 4-substituent groups from different directions (as shown in Figure 10c,d). The side chain carboxyl group of Asp458 lay between the 4-imino and terminal imino groups (in the complex 5n, the terminal amino group) and probably interacted with both groups. However, Asp458 was closer to the terminal amino/imino group and might interact more strongly with it. The average distances between the carboxyl carbon and the hydrogen on the terminal amino/imino group of Asp458 were 0.882 nm (5n), 0.744 nm (5h), 0.715 nm (5g), and 0.723 nm (5e), respectively. The order was qualitatively consistent with the E_MM_ contributions of the residues in the four complexes, i.e., the shorter the distance between the charged atoms, the greater the E_MM_ contributions. The side chain carboxyl group of Asp444 only interacted with the terminal amino or imino groups, so it had a smaller E_MM_ contribution than Asp458 in the other three complexes, except complex 5n. The characteristic terminal amino group in complex 5n was closer to the carboxyl group of Asp444, and therefore the E_MM_ contribution was greater than in the other three complexes. Wang et al. also found that these two residues formed salt bridges with inhibitors in complexes 5n and 5e, and suggested that this might be the primary role of inhibitors binding to proteins [26]. However, the binding free energy calculations that we performed indicated that in complex 5e, the E_MM_ of Asp444 was small and only a minor interaction.

The above results indicate that the proteins formed more electrostatic interactions with the 4-substituent groups, but these electrostatic interactions were all relatively weak compared to those in the hinge region and were not sufficient to fix the conformations of the 4-substituent groups. In addition, the electrostatic interactions had high directional requirements, and the conformations of these electrostatically interacting residues in different complexes were different to some extent. The above factors resulted in easy rotations of 4-substituents in the different complexes and, in turn, influenced the binding free energy of their surrounding residues.

### 2.9. The Factors Affecting the Inhibitory Capacity of Different Inhibitors

As mentioned above, the four inhibitors we studied were all competitive inhibitors, and their inhibitory capabilities were mainly related to their enzyme-inhibitor affinities. However, the four inhibitors had the same parent nucleus and very similar 2-substituent and 4-substituent groups, with up to 1000-fold or more differences in inhibitory capacities between them. The reasons for the differences in their inhibitory capacities were discussed in our subgroups.

The inhibitors 5n and 5h had the same 2-substituent groups, differing only in the terminal amino/imino group of the 4-substituent groups. The residues Ile327, Lys350, Glu366, Asp405, Asp444, and Asp458 all formed electrostatic interactions with the two amino groups on the 4-substituent. There was a terminal amino group in inhibitor 5n, and all residues, except Glu366, interacted more strongly with this terminal amino group. Therefore, inhibitor 5n had a higher affinity for the enzyme than inhibitor 5h. By docking, Wang et al. concluded that Ala402 had a different interaction with the Cl atom on the 2-benzene ring in inhibitor 5n than other inhibitors, which was the main reason why inhibitor 5n inhibited more than other inhibitors [26]. However, in our work, both conformational observation and hydrogen bonding statistics and binding free energy analysis indicated that there was no significant interaction between Ala402 and the inhibitor 5n.

The inhibitors 5h, 5g, and 5e had the same 4-substituents, differing only in halogen atoms on the 2-benzene rings. Our research showed that the differences in halogen atoms could cause two impacts. First of all, the conformation of Ile327, which had a hydrophobic interaction with it, could be greatly altered, affecting the position of the 2-benzene ring. Additionally, as discussed above, differences in halogens might cause differences in the flip of the 4-imino group, which could also have a secondary effect on the position of the parent nucleus. The combined result of these two effects resulted in some differences in the binding positions of the four inhibitors in the enzymes, particularly the 2-substituent and the parent nucleus (as shown in Figure 13, the 2-benzene ring positions of inhibitors 5n and 5h with the same halogens were basically the same, while 5g and 5e with different halogens were different), and caused the E_MM_ contributions of the hinge region residues Glu396, Leu398, Glu399, and Gly401, which interacted with the inhibitors, to be different in the four complexes. Of these, complex 5e had the most variable inhibitor positions and the weakest E_MM_ contributions of Glu396, Leu398, and Glu399. The electrostatic interactions of residues such as Lys350, Glu366, Asp405, Arg589, and Arg591 with the 4-imino were also affected by the flip, but these electrostatic interactions became either stronger or weaker and had a much smaller effect on inhibitor and protein affinities than the effect of changes in the hinge region. 

## 3. Materials and Methods

### 3.1. Molecular Docking Calculations

The docking operation was performed by using the AutoDock Vina software [33,34,35]. The key residues mentioned in the Introduction that bind to ATP were all contained in the box. The box size was set to 16 × 20 × 24, and the grid spacing was set to 1 Å. The number of conformations in the docking output was set to 20. The docking results with low energy and similar binding positions and binding modes to those reported in the literature were selected.

### 3.2. Molecular Dynamics Simulations

The molecular dynamics simulations were performed using the GROMACS 2019.6 program (Emile Aplo et al., Stockholm, Sweden) [36,37,38]. The AMBER99SB-ILDN force field was applied to PAK4 [42]. The GAFF field [43] of the inhibitors was generated by the antechamber program in the AMBER16 package [44]. Next, the entire complex systems were solvated using the TIP3P water molecule model [45]. The shortest distance of the protein edge from the boundary of the cubic solvent box was 10Å. To maintain the electrical neutrality of the complex systems, three chloride ions were added to the system to neutralize the positive charge in the protein. The steepest descent method was used to minimize the energy of the system in 1000 steps to ensure that there were no atom collisions or unreasonable spatial structures. Two stages of equilibration were then carried out, the first with an NVT equilibration of 500 ps at 300 K and the second with an NPT equilibration of 500 ps at the same temperature with the pressure set at 1 atm. The final product simulations were carried out separately for 200 ns. The integration step was set to 2 fs, and the long-range electrostatic cut-off radius was 12 Å. Save coordinates and energy information every 10 ps. Two sets of parallel simulations were carried out for each system to ensure the reliability of the results. 

### 3.3. Bonding Free Energy Calculations

Binding free energy is an important metric for determining receptor–ligand affinity [46]. The molecular mechanics Poisson–Boltzmann surface area (MM/PBSA) method is an effective method for calculating binding free energies and was widely used in the calculation of biomolecular interactions because of its efficiency and low computational cost [47]. The binding free energy of the complex system in the solvent is defined as follows [48]: ΔG_bind_ = ⟨G_complex_⟩ − ⟨G_protein_⟩ − ⟨G_ligand_⟩(1)
where G_complex_ represents the free energy of the complex in the solvent, G_protein_ and G_ligand_ represent the free energy of the protein and ligand alone in the solvent, respectively, and the pointed brackets represent the average value. The free energy of each of these monomers can be expressed as:⟨G_x_⟩ = ⟨E_MM_⟩ − ⟨TS⟩ + ⟨G_solvation_⟩(2)

G_x_ represents the free energy of system x, which is one of the complexes, proteins, and ligands. The free energy of the monomer is the sum of the free energy of the vacuum and the free energy of solvation. E_MM_ represents the potential energy of molecules in a vacuum; TS represents the entropic contribution to the free energy in a vacuum [48]; T and S represent temperature and entropy, respectively; G_solvation_ represents the free energy of solvation. E_MM_ in a vacuum consists of bonded interaction (E_bonded_) and non-bonded interaction (E_nonbonded_). The bonded interactions (also called internal energy) include the bond lengths, bond angles, and dihedral angles; the non-bonded interactions consist of electrostatic and van der Waals interactions, which are modeled using the Coulomb and Lennard–Jones (LJ) potential functions, respectively [49].
E_MM_ = E_bonded_ + E_nonbonded_ = (E_bond_ + E_angle_ + E_dihedral_) + (E_elec_ + E_vdW_)(3)

The free energy of solvation consists of two components:G_solvation_ = G_polar_ + G_nonpolar_(4)
where polar and non-polar are the electrostatic and non-electrostatic contributions to the free energy of solvation, respectively.

## 4. Conclusions

The binding modes and inhibitory mechanisms of four 7*H*-pyrrolo[2,3-d]pyrimidine derivatives as competitive inhibitors of p21-activated kinase 4 were studied in this paper. The results show that the main binding regions of enzymes and inhibitors were the hinge region, β1–β3 and β7 sheets, and residues with charged side chains around the 4-substituent of inhibitors. Of these, the hinge region interacted most strongly with the inhibitors of all the complexes and was the main reason for their inhibitory capacities, but not for the differences in inhibitory capacities of the inhibitors. Unlike the other three, the terminal amino group of inhibitor 5n promoted enhanced interactions with the inhibitor of residues such as Ile327, Lys350, Asp405, Asp444, and Asp458, which formed hydrogen bonds or electrostatic interactions with the inhibitor, increasing the stability of the inhibitor in the enzyme and providing the strongest inhibitory capacity in the experiment. The different 2-substituents of the inhibitors 5h, 5g, and 5e caused differences in the binding free energy of Ile327 and in the 2-benzene ring binding position, which further affected the hinge region interactions. The combined effect of these two resulted in the weakest interaction of inhibitor 5e with the protein and the weakest inhibitory capacity measured in the experiments. There was also a difference in the 2-benzene ring position between inhibitor 5h and 5g, which resulted in stronger interaction between inhibitor 5h and Met395 and Gly401 in the hinge region. More importantly, inhibitor 5h led to enhanced electrostatic interactions with charged residues such as Lys350, Glu366, Asp405, Arg589, and Arg591 due to the flipping of the 4-imino group. Therefore, the enzyme-inhibitor interaction in complex 5h was stronger than that in complex 5g, but the larger solvation energy difference between the two offsets some of this, resulting in little difference in total binding free energy between the two complexes.

## Figures and Tables

**Figure 1 molecules-28-00413-f001:**
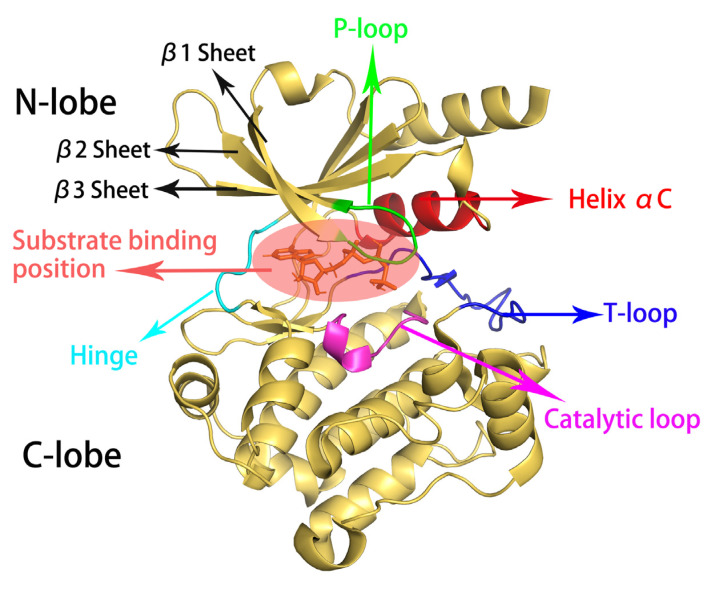
The cartoon representation of the kinase structural domain of PAK4. The P-loop is shown in green, Helix αC in red, Hinge in cyan, catalytic loop in pink, and T-loop in blue; the binding site of ATP is circled by a light red ellipse, and β1-β3 are indicated by arrows, respectively.

**Figure 2 molecules-28-00413-f002:**
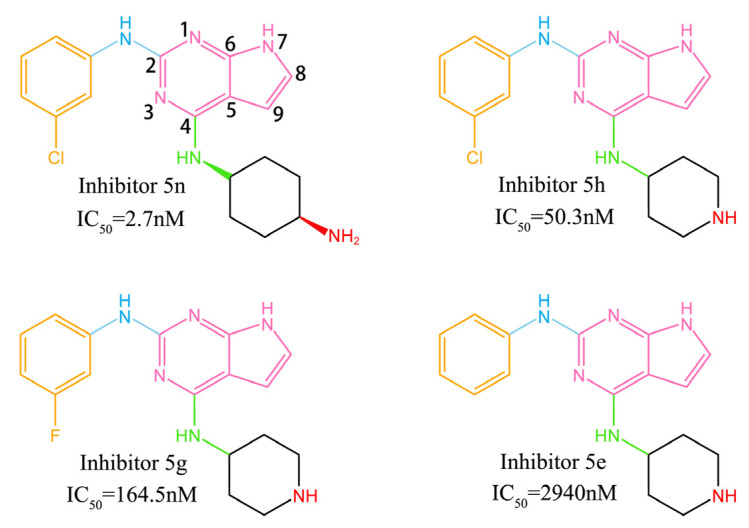
The 2D structures of four pyrrolo [2,3-d] pyrimidine inhibitors. The atomic number of the parent nucleus is marked on the inhibitor 5n. The different colors represent the different parts of the inhibitors. The pink group is the pyrrolo [2,3-d] pyrimidine parent nucleus; the orange group is called the 2-benzene ring; the blue group is called the 2-imino; the green group is called the 4-imino; and the black group is called the 4-hexatomic ring. The red group is called the terminal amino group (inhibitor 5n) or the terminal imino group (inhibitor 5h, 5g, and 5e).

**Figure 3 molecules-28-00413-f003:**
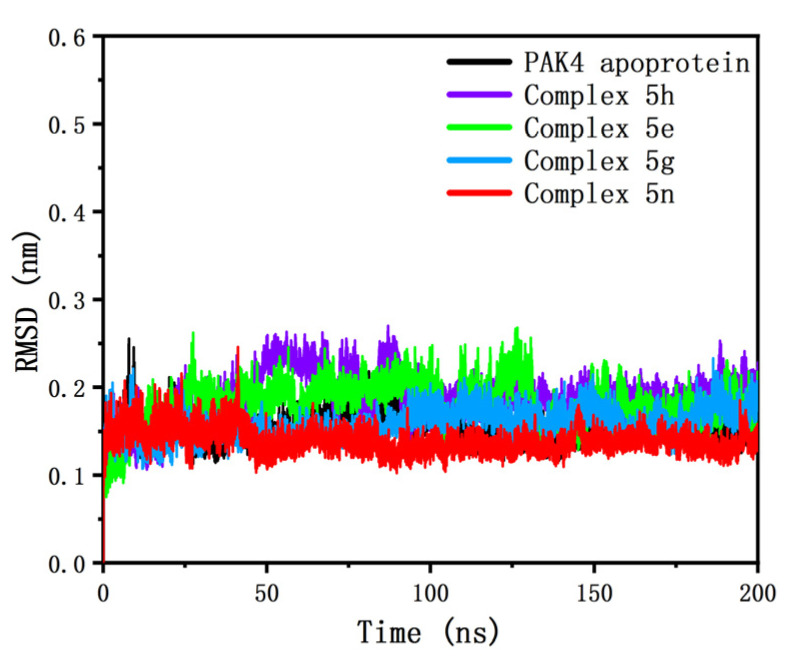
The RMSD of the Cα atomic backbone of five systems as functions of time.

**Figure 4 molecules-28-00413-f004:**
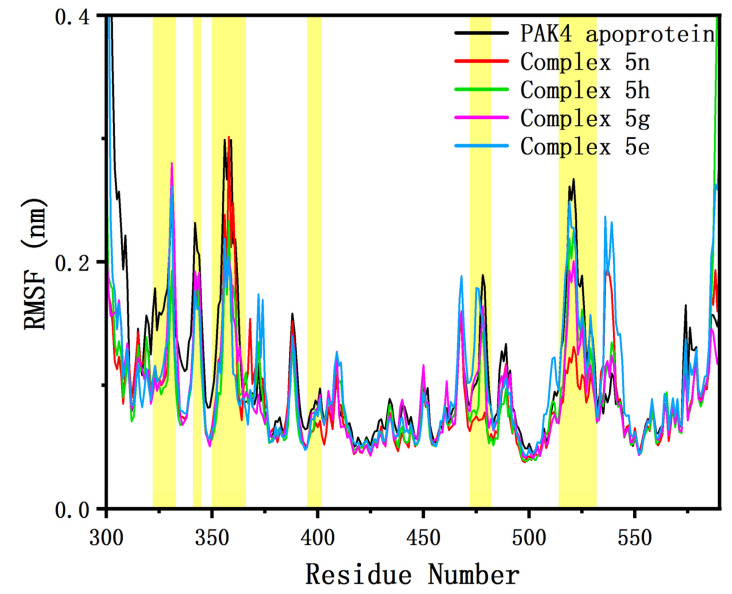
The comparisons of the RMSF of four complexes with the PAK4 apoprotein. The regions marked in yellow background are, in order: Asp322-Gly333, Arg341-Lys345, Lys350-Glu366, Met395-Ala402, Arg472-Met482, and Pro514-Ser532.

**Figure 5 molecules-28-00413-f005:**
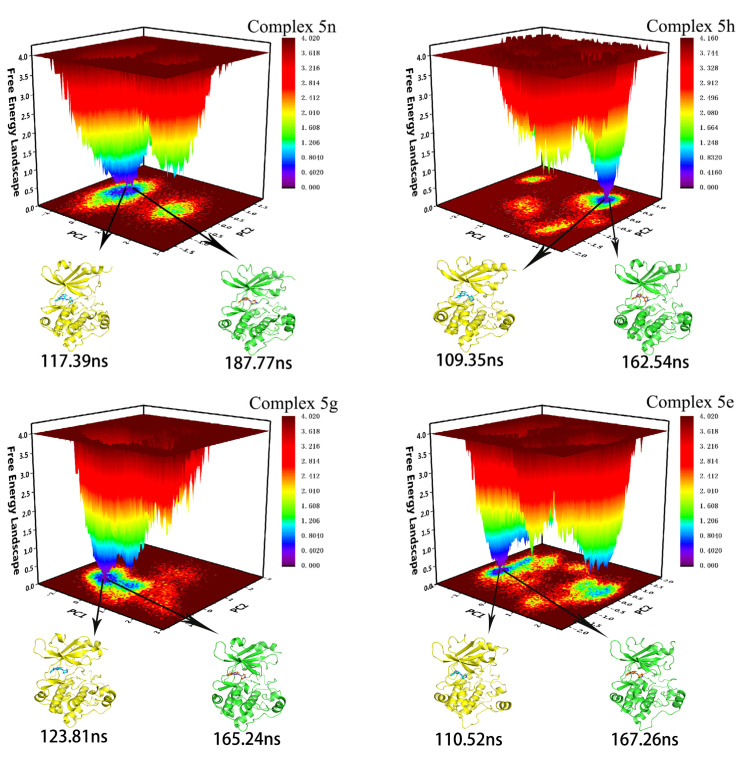
The free energy landscapes of four complex systems and the samples extracted from the MD simulation trajectories.

**Figure 6 molecules-28-00413-f006:**
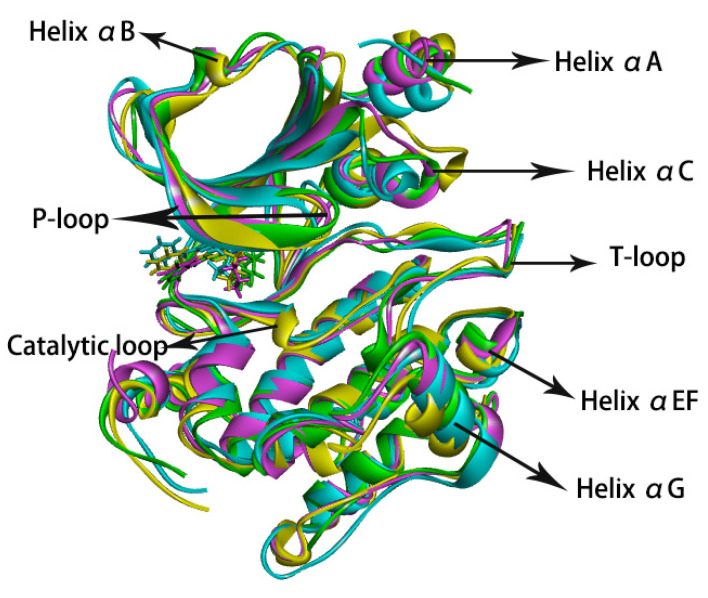
The conformational overlap of four complexes. Complex 5n is shown as green, Complex 5h is shown as pink, Complex 5g is shown as yellow, and Complex 5e is shown as cyan.

**Figure 7 molecules-28-00413-f007:**
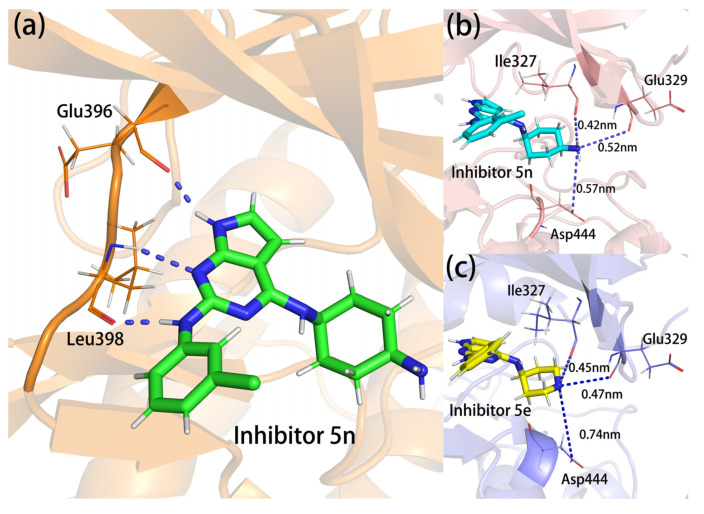
(**a**) The structure of the triple hydrogen bonds in the hinge region; (**b**,**c**) the distances between Ile327, Glu329, Asp444, and inhibitors 5n and 5e.

**Figure 8 molecules-28-00413-f008:**
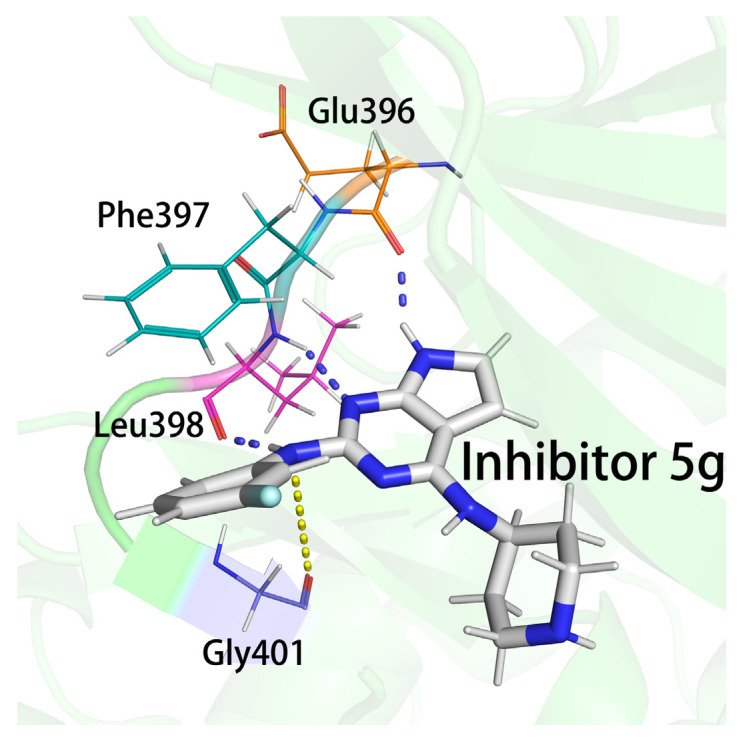
The locations and interactions of hinge region residues with inhibitors. The formation of triple hydrogen bonds between Glu396 and Leu398 and the inhibitor is indicated by the blue dashed line, and the electrostatic interaction of the oxygen atom on the Gly401 backbone with the 2-imino group is indicated by the yellow dashed line.

**Figure 9 molecules-28-00413-f009:**
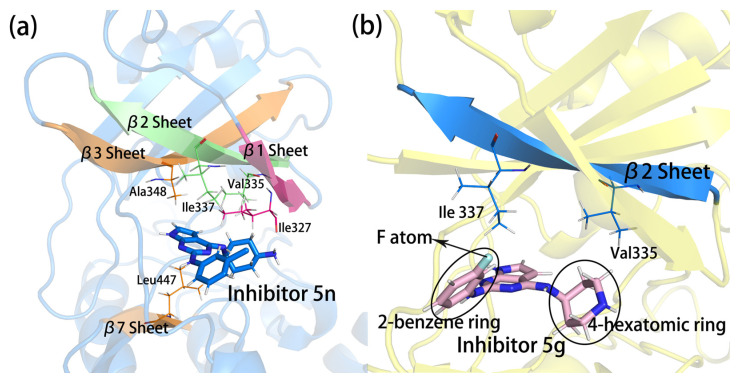
The positions of residues on the β-sheets in relation to the inhibitors. (**a**) Residues on the β-sheets with strong interactions with the inhibitors; (**b**) position relationships of Val335 and Ile337 with the 2-benzene ring and 4-hexamoric ring.

**Figure 10 molecules-28-00413-f010:**
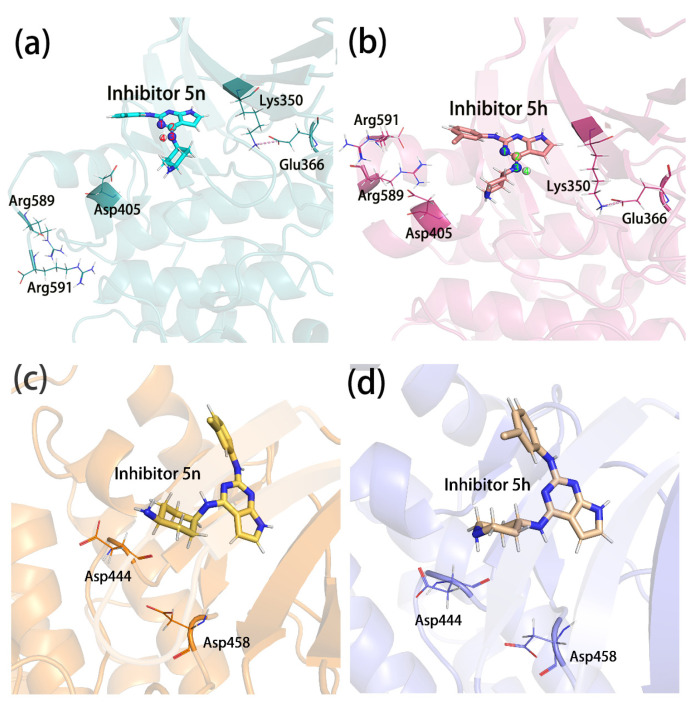
(**a**–**d**) The relationships of the charged residues to the 4-substituent position of the inhibitors. The salt bridges formed by Lys350 and Glu366 in (**a**,**b**) are shown as pink dashed lines, and the four atoms forming the dihedral angles are shown as balls and sticks and marked as 1, 2, 3, and 4, respectively.

**Figure 11 molecules-28-00413-f011:**
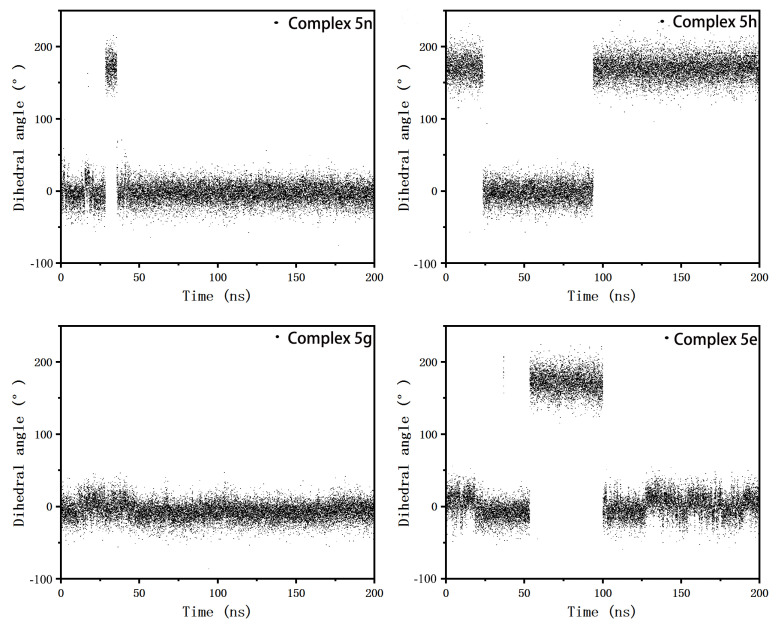
The variations of the dihedral angles reflecting the 4-imino directions in four complex systems with simulation time.

**Figure 12 molecules-28-00413-f012:**
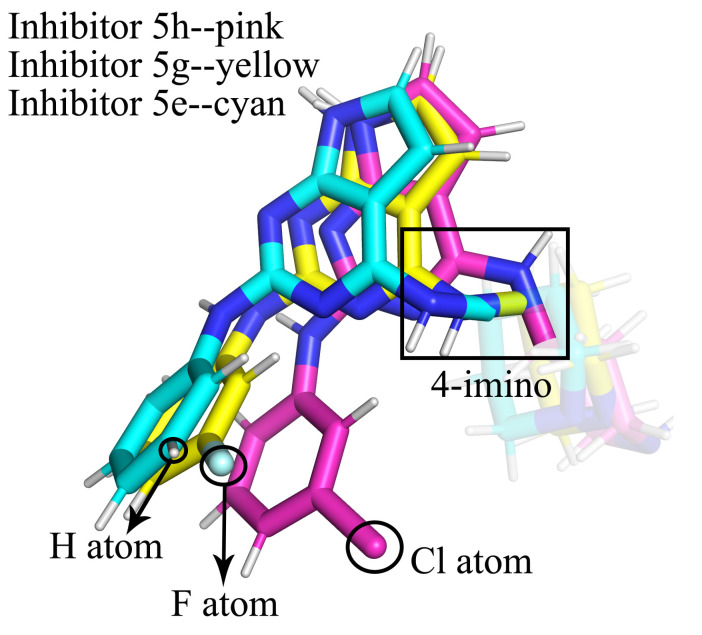
The positions of the halogen atoms on the 2-benzene rings in relation to the 4-imino groups in inhibitors 5h, 5g, and 5e.

**Figure 13 molecules-28-00413-f013:**
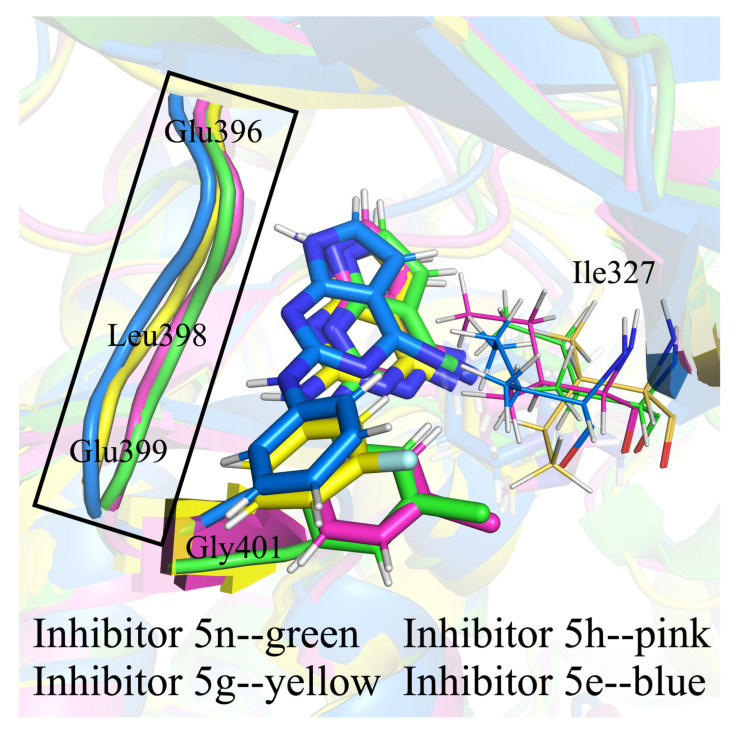
The conformational comparison of Ile327, the 2-benzene ring, and the hinge region of the inhibitors in the four complexes.

**Table 1 molecules-28-00413-t001:** The E_MM_ contributions of important residues (kJ·mol^−1^).

Residue	Complex 5n	Complex 5h	Complex 5g	Complex 5e
Glu396 Phe397 Leu398 Gly401	−17.0917 −13.9827 −21.0516 −8.1326	−18.4948 −14.1506 −21.0612 −7.7494	−18.2980 −13.7404 −20.2117 −4.0624	−14.6707 −13.3797 −19.1500 −7.6172
Ile327 Val335 Ile337 Ala348 Leu447	−10.8861 −4.0020 −2.2354 −3.3588 −9.1495	−8.6148 −3.4731 −2.0533 −4.0180 −8.9728	−7.6158 −3.0272 −6.4054 −4.0549 −6.8793	−8.2873 −4.4933 −2.4832 −3.6271 −7.4401
Lys350 Glu366 Asp405 Asp444 Asp458 Arg589 Arg591	3.6187 −3.8704 3.0469 −7.8068 −5.5331 −0.1275 −0.0779	6.8103 −6.0667 6.3600 −1.3075 −6.7903 −4.2941 −5.0457	4.7075 −3.7092 1.9215 −5.3929 −8.6775 0.0255 0.0685	4.9280 −3.9110 2.7995 −1.4685 −6.8339 −0.4139 −0.3159

## Data Availability

Not applicable.

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
