# Peer review of "The Inhibitory Mechanism of 7H-Pyrrolo[2,3-d]pyrimidine Derivatives as Inhibitors of P21-Activated Kinase 4 through Molecular Dynamics Simulation"

_molecules, 2023, doi:10.3390/molecules28010413_

Round 1
Reviewer 1 Report
The manuscript of Zhang and coworkers concerns a very accurate analysis of results arising from a computational study, devoted to comparing inhibitors of PAK 4 enzyme and to provide atomistic insights about their binding to the protein. Overexpression of PAK 4 has been linked to the onset of a number of cancers and for this reason its inhibition is potentially considered for new therapeutic strategies.
The protocol adopted by the authors represents the state of the art molecular modeling and it has been coherently/competently presented, carried out and described. In addition, I appreciated the declared will of the authors to contribute in the providing of additional molecular information that might be useful in rational drug design. For all these reasons, and considering that the article fits well with the special issue, I support the publication in Molecules.
I only have few comments/suggestions for the authors that, according to my point of view, can increase the readability of the manuscript for the broader readers-community of the journal, as follows:
- In the introduction, line 85-86, the authors say that the “The protein in docking is rigid”; I understand that here they are referring to the docking calculations carried out in a previous work, but, written like that, it seems that they attest that all docking simulations are rigid, which is not. I would rephrase this sentence, to avoid misunderstanding the readers.
- I wonder if the authors can provide a text file for supporting information, including some technical tables and graphs that might be interesting only for experts of molecular simulations. Obviously, it is a matter of personal taste but I would include in such file Figure 7, Table 1, 2 and 3. My opinion is that lightening the article from technical details can increase the overall readability of the manuscript, making it interesting also for those who are not experts of the field.
- I would make one graph of superimposed lines in the case of Figure 4 and Figure 12, instead of the four separated ones. In this way, the comparison is immediate and it can follow easily the written description.
- Since there is not a direct comparison with experiments that provide kinetic info about the binding of the considered ligands, I would avoid the discussion of obtained ΔG in term of absolute values; I would present these results in the text mention the relative ΔΔG obtained, highlighting that the referring zero has been selected in accordance to the lowest calculated ΔG.
- I'm not a fan of description of results in the form of lists. My opinion is that the authors can improve the sections from line 154to175 and from line 285to341, adjusting these sentences within the good flow of the manuscript.
Reviewer 3 Report
In this study, Du et al caulates the binding affinity of PAK4 inhibitors with PAK4 using molecular docking, molecular dynamics simulation, and binding free energy calculation. The authors discuss the factors and their relative importance that influence the binding infinity, including hydrogen bonding, electrostatic interaction, vdw forces and the corresponding conformation environments .
The method is reasonable, the results are rich and discussions are in detailed. The paper can be published after the following comments are addressed:
1. The motivation and significance of this study is not clear. The authors spent lengthy introduction on the PAK4 and related background, but very little on the significance and the rationale of pyrrolo [2,3-d] pyrimidine, which is the focus of this paper. In particular, why do we need to study the biding affinity of the four molecules?
2. line 96-102, the definition of the two comparison groups is not clear. These sentences are confused and it is not clear what's the comparison.
3. section 2.3: How is RMSD related to free energy and conformation changes? The author may want to give more explanations and theoretical background.
4. The authors mention the binding affinity can differs for 1000 fold of different substitute in experiments. This is not observed in this simulation study. The authors need to explain the possible reasons for the discrepancy.
5. What's the influence and importance of the solvation free energy changes before and after binding?
Round 2
Reviewer 3 Report
The authors have well addressed my concerns. Thus, I recommend publication.